# Recent Developments and Perspectives of Cobalt Sulfide-Based Composite Materials in Photocatalysis

Hui-Qi Chen, Jin-Ge Hao, Yu Wei, Wei-Ya Huang ⬡, Jia-Lin Zhang ⬡, Tao Deng, Kai Yang ⬡ and Kang-Qiang Lu *⬡

Jiangxi Provincial Key Laboratory of Functional Molecular Materials Chemistry, School of Chemistry and Chemical Engineering, Jiangxi University of Science and Technology, Ganzhou 341000, China

* Correspondence: kqlu@jxust.edu.cn

**Abstract:** Photocatalysis, as an inexpensive and safe technology to convert solar energy, is essential for the efficient utilization of sustainable renewable energy sources. Earth-abundant cobalt sulfide-based composites have generated great interest in the field of solar fuel conversion because of their cheap, diverse structures and facile preparation. Over the past 10 years, the number of reports on cobalt sulfide-based photocatalysts has increased year by year, and more than 500 publications on the application of cobalt sulfide groups in photocatalysis can be found in the last three years. In this review, we initially summarize the four common strategies for preparing cobalt sulfide-based composite materials. Then, the multiple roles of cobalt sulfide-based cocatalysts in photocatalysis have been discussed. After that, we present the latest progress of cobalt sulfide in four fields of photocatalysis application, including photocatalytic hydrogen production, carbon dioxide reduction, nitrogen fixation, and photocatalytic degradation of pollutants. Finally, the development prospects and challenges of cobalt sulfide-based photocatalysts are discussed. This review is expected to provide useful reference for the construction of high-performance cobalt sulfide-based composite photocatalytic materials for sustainable solar-chemical energy conversion.

**Keywords:** photocatalysis; cobalt sulfide; synthesis strategies; multiple roles; energy conversion

## 1. Introduction

Energy has been the primary driving force behind civilization throughout human history. At the moment, the heavy reliance on fossil fuels has led to severe global problems such as an energy crisis and pollution of the environment [1,2]. Photocatalytic technology is an effective approach for the photochemical conversion and storage of solar energy [3–7]. Over the past 20 years, exploring new photocatalyst materials and their reaction mechanisms has been a top priority [8,9]. A complete photocatalytic reaction includes the following three steps: (i) light absorption; (ii) separation and migration of the photogenerated charges; and (iii) surface reduction and oxidation reactions [10,11]. Various photocatalysts, such as metal-free [12], metal oxides [13], metal sulfides [14], metal phosphides [15], and metal selenides [16], have been extensively studied in previous reports. However, most single-component photocatalysts have shown unsatisfactory photocatalytic activity due to weak spectral absorption, fast recombination of the photogenerated charge carrier, and an insufficient active site.

To address these problems, loading suitable cocatalysts is considered an effective way to facilitate photocatalytic reactions. Nowadays, many precious metals have been developed as cocatalysts, such as gold, platinum, and palladium [17,18]. In spite of this, the scarcity and expensiveness of these noble metals severely restrict their use on a large scale. In recent years, transition metal cocatalysts have received much attention in the field of photocatalysis due to their advantages of low cost, an abundance resources, good stability, and high catalytic activity. Especially, noble-metal-free cobalt sulfide ($Co_xS_y$)

has been widely explored for substituting the noble metal cocatalyst due to its sufficient catalytic sites, multivalent states, and diverse structures [19–22]. Furthermore, several excellent papers have discussed the work of cobalt sulfide-based cocatalysts in enhancing the photocatalytic properties. For example, Zhu et al. have reported a novel study on the application of cobalt sulfide-modified graphite carbon nitride for photocatalytic hydrogen evolution [23]. Cobalt sulfide acts as a cocatalyst to promote the migration of excited electrons from graphite carbon nitride to cobalt sulfide. In addition, Xu et al. also investigated the photocatalytic properties of $Co_3S_4$/$Ag_2S$ nanocomposite, and the photocatalytic performance of binary nanomaterials was higher than that of a single catalyst structure [24]. In addition, Kokilavani et al. have proposed a facile chemical precipitation method for the synthesis of CoS/$Ag_2WO_4$ photocatalyst, and the composites show excellent performance for photocatalytic degradation and antibacterial activity [25].

Although the application of cobalt-based photocatalysts has been reviewed in the previous literature [26], there are few specific reviews that systematically summarize the synthesis, multifaceted roles, and various applications of cobalt sulfide-based composite photocatalytic materials. Therefore, it is necessary to conduct a comprehensive review of the current research progress of cobalt sulfide-based composite photocatalysts to expand their practical applications. As shown in Figure 1, in this review, we have elaborated on the synthesis methods of the cobalt sulfide-based composites. Then, the roles of cobalt sulfide-based cocatalysts in photocatalysis have been discussed. Furthermore, recent advances of the cobalt sulfide-based composite photocatalysts in photocatalytic $H_2$ production, $CO_2$ reduction, nitrogen fixation, and pollutant degradation are also reviewed. Finally, we put forward the unsolved problems and possible future development directions of cobalt sulfide-based composites in a variety of photocatalytic applications. It is hoped that this review could provide useful information for rationally designing high-performance cobalt sulfide-based composite photocatalysts.

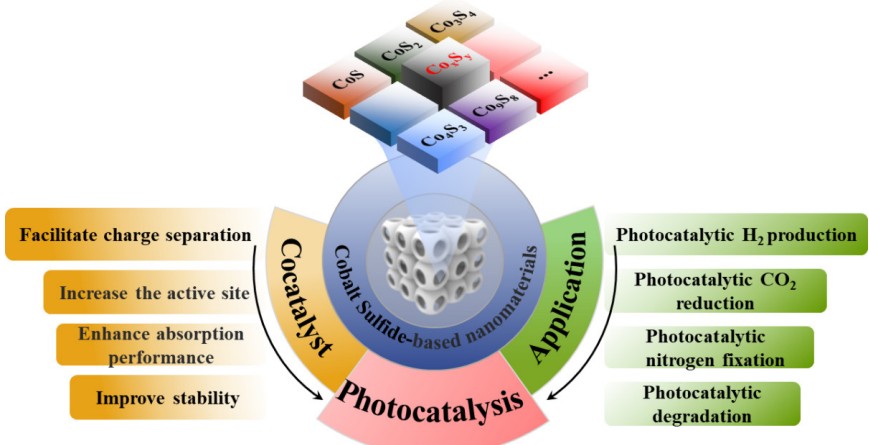

**Figure 1.** Schematic illustration of multifarious roles and applications of $Co_xS_y$ in heterogeneous photocatalysis.

## 2. Synthesis of Cobalt Sulfide-Based Composite Material

The photocatalytic activity of cobalt sulfide-based photocatalysts mainly depends upon their size, morphology, specific surface area, and crystal structure [27,28]. Up to now, various phase and morphological cobalt sulfide-based species have been synthesized for photocatalysts. Here, in this section, we focus on the strategies for the synthesis of cobalt sulfide-based hybrid materials.

### 2.1. Wet Chemistry Method

For the synthesis of cobalt sulfide-based composite photocatalysts, wet chemistry has proven to be an effective technique with thioacetamide (TAA) and thiourea (TU)

as S sources [29,30]. Briefly, the vulcanization of different cobalt precursors (such as cobalt chloride or cobalt nitrate) could be achieved by using thioacetamide or thiourea in hot organic solvents at inert temperatures. In most cases, this method can be used to synthesize uniformly sized cobalt sulfide-based composite photocatalysts. As mentioned in the previously reported literatures, cobalt sulfide-based catalysis topography is mainly controlled by the S/Co ratio, solvent selection, and reaction conditions. For example, Qian et al. have synthesized $Co_3S_4$ by using cobalt nitrate and thioacetamide as cobalt and sulfur sources [31]. As shown in Figure 2a, $Co_3S_4$ showed a hollow dodecahedral structure with an average particle size of about 800 nm. In addition, the temperature has an obvious effect on the catalyst morphology. For instance, Qiu et al. have used the same cobalt-based precursor material and thiourea but different heating temperature. However, as shown in Figure 2b, a flower-like morphology of $Co_3S_4$ has been obtained [32]. As shown in Figure 2c,d, the flower-like morphology of $Co_3S_4$ can also package $MoS_2$ to build a nuclear-shell heterojunction photocatalyst, which shows excellent performance in the photocatalytic degradation pollutant. When constructing cobalt sulfide-based composites with specific morphology, structure, and size, it is advantageous to use wet chemistry methods, and thus wet chemistry synthesis is more common than gas-solid synthesis in the syntheses of cobalt sulfide-based composite photocatalysts [33].

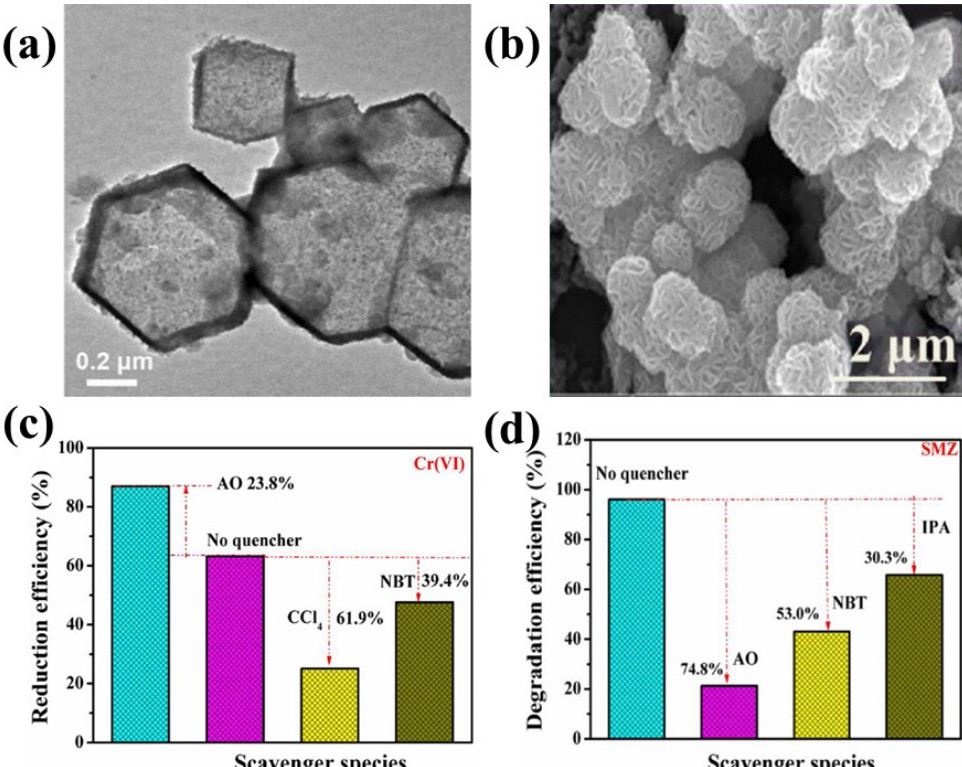

**Figure 2.** (**a**) TEM image of $Co_3S_4$. Reprinted with permission from ref. [31]. Copyright 2022, Elsevier. (**b**) SEM image of $Co_3S_4$. (**c**,**d**) Active species trapping experiments for (**c**) reduction of Cr(VI) and (**d**) degradation of SMZ. Reprinted with permission from ref. [32]. Copyright 2020 Elsevier.

## 2.2. Gas-Solid Method

Although wet chemistry techniques can produce cobalt sulfide with well-defined nanostructures, the complicated processes and low yields prevent their widespread applications [34]. Recently, the gas-solid synthesis of cobalt sulfide has attracted a wide range of attention. For the gas-solid method, sulfur powder and the cobalt-based precursor are placed in two boats of porcelain, respectively, with sulfur powder located upstream of the furnace. Following that, the samples are heated in a static atmosphere at a prescribed temperature. At temperatures over 450 °C, sulfur powder decomposes to release $H_2S$, which

reacts with cobalt-based precursors to form cobalt sulfide. Under certain temperatures, this surfactant-free method allows the morphology of the precursors to be preserved well, which makes it more applicable for developing different 3D self-standing cobalt sulfides with different structures [35]. For example, Xie and his collaborators synthesized the Graphdiyne-CoS$_2$ heterojunction nanocomposites by placing the Co(OH)F/CC composite in the tubular furnace at 500 °C [36]. Moreover, as shown in Figure 3b, Wang et al. have reported the synthesis of cobalt sulfide with multi-shell nanobox morphology by annealing the cobalt-based MOF precursor at 350 °C [37]. Cobalt sulfide with different shell numbers can be obtained by adjusting the number of shells in the cobalt-based MOF precursor, and the performance of three-layer shells is the best.

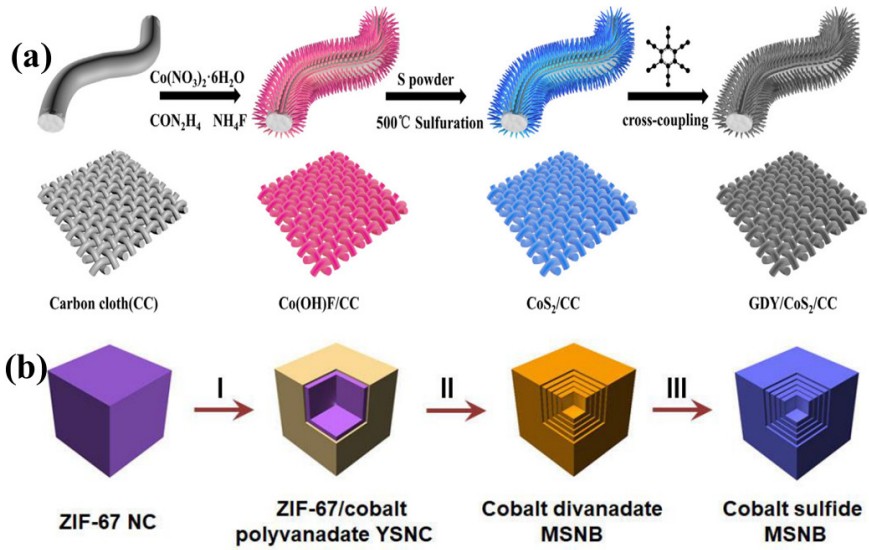

**Figure 3.** (**a**) A schematic diagram of the phase synthesis of GDY/CoS$_2$/CC catalyst. Reprinted with permission from ref. [36]. Copyright 2021, Elsevier. (**b**) A schematic illustration of the formation process of cobalt sulfide MSNB. Reprinted with permission from ref. [37]. Copyright 2019, Wiley.

### 2.3. Photo-Deposition Method

In many reports, the photo-deposition-assisted cocatalysts have higher catalytic performance than conventional methods because photo-deposition can promote the deposition of cocatalysts on semiconductors with well-matched positions and provide greater contact area and more active sites, thereby accelerating interfacial charge separation between the semiconductor and the co-catalysts [38,39]. The photo-deposition method was applied to deposit CoS$_2$ nanoparticles on g-C$_3$N$_4$ as reported by Yang et al. [38]. Appropriately sized CoS$_2$ nanoparticles have high adsorption and photocatalytic hydrogen production performances. The experimental results further show that the electron aggregation ability of the cocatalyst is based on the size effect of CoS$_2$, and the appropriate size of the cocatalyst can effectively promote the separation of photogenerated electron-hole pairs. Due to its simple and time-saving operation and good photocatalytic activity, the photo-deposition method can be used to realize the development of new non-noble metal photocatalytic materials. Moreover, amorphous cobalt sulfide obtained by the photo-deposition method has been used as an effective cocatalyst for photocatalytic water decomposition properties. For example, as exhibited in Figure 4a, Chen et al. have successfully loaded amorphous CoS$_x$ nanodot cocatalyst onto rGO nanosheets through the photo-deposition strategies [40]. Incorporation of amorphous cobalt sulfide nanodot can significantly improve the catalytic activity of rGO/TiO$_2$ photocatalysts based on the formation of new active sites (Figure 4b,c). Compared to the crystalline phase, the amorphous CoS$_x$ structure can effectively inhibit electron-hole recombination and provide a large number of active sites, exhibiting higher hydrogen production activity, thus accelerating the transfer of electrons and improving the surface H$_2$ precipitation rate.

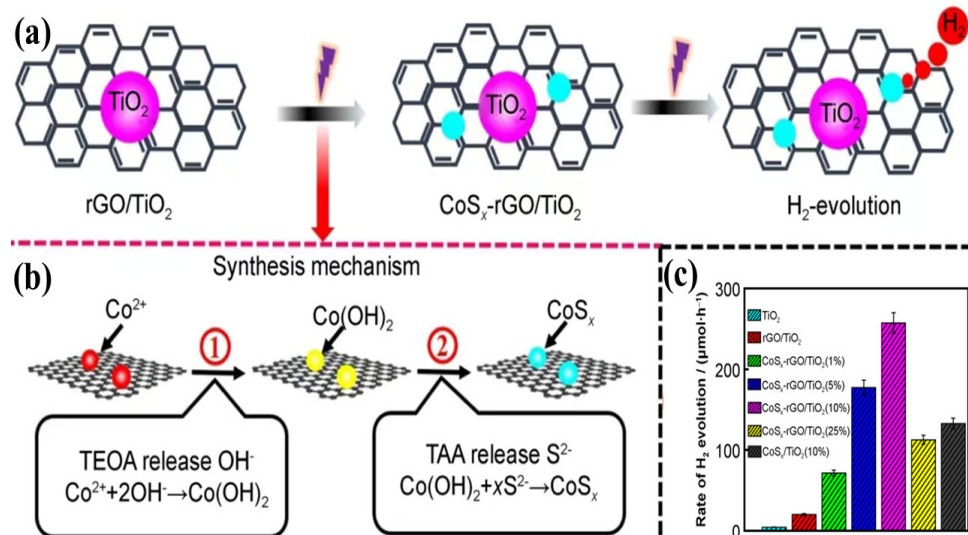

**Figure 4.** (**a**) Graphical illustration for synthetic process of CoS$_x$-rGO/TiO$_2$. (**b**) Formation mechanism of CoS$_x$ on rGO surface. (**c**) H$_2$-production rate of TiO$_2$, rGO/TiO$_2$, CoS$_x$-rGO/TiO$_2$ (1%), CoS$_x$-rGO/TiO$_2$ (5%), CoS$_x$-rGO/TiO$_2$ (10%), CoS$_x$-rGO/TiO$_2$ (25%), and CoS$_x$/TiO$_2$ (10%). Reprinted with permission from ref. [40]. Copyright, 2021 Springer Nature.

### 2.4. Electrochemical Reaction Method

Several electrochemical methods, including pulse electrochemical reduction and anodic oxidation, have been reported for the synthesis of cobalt sulfide-based cocatalysts. As demonstrated in Figure 5, a facile and inexpensive one-step anodization method has been developed by Bian et al. to synthesize cobalt sulfide (CoS$_x$) nanosheets with mesoporous structures [41]. This porous, reverse-porous, self-grown nanostructure provides high surface-active sites for catalytic reactions and facilitates electron transfer between active materials, exhibiting excellent hydrogen evolution (HER) and oxygen evolution (OER) performances. In addition, Kubendhiran et al. synthesized the cobalt sulfide/reduced graphene oxide (CoS/rGO) nanohybrid using a single-step electrochemical method. The obtained CoS/rGO nanocomplexes show excellent selectivity and catalytic activity for H$_2$O$_2$ [42].

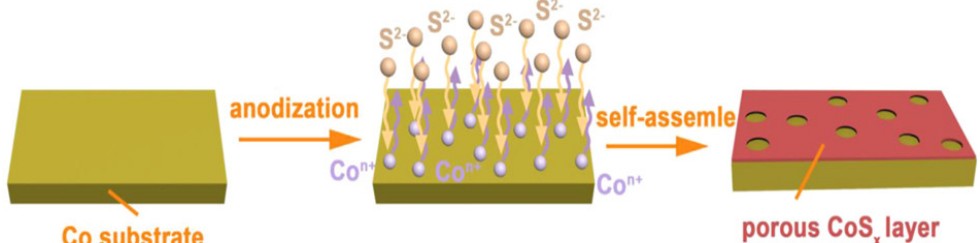

**Figure 5.** Schematic illustration of the anodization process of CoS$_x$ on the metal substrate. Reprinted with permission from ref. [41]. Copyright, 2021 Elsevier.

### 2.5. Other Methods

In addition to the several most commonly used methods mentioned above, chemical vapor deposition (CVD) [43], microwave-assisted methods [44,45], and template-assisted methods [46] have also been used to prepare cobalt sulfide-based hybrid materials. The CVD method often requires a cumbersome manufacturing process or expensive equipment, and the prepared cobalt sulfide-based composites have poor water solubility. Therefore, CVD methods are rarely used to synthesize cobalt sulfide-based composite photocatalysts. In addition, compared with the traditional hydrothermal method, the microwave method can significantly shorten the reaction time. For instance, Souleymen et al. have synthesized

graphene-based cobalt sulfide freestanding sheets with microwave assistance [45]. The $CoS_x$ non-layered and freestanding nanosheets were formed and exhibited higher catalytic activity due to their thin thickness, large surface area, and abundant pores compared with layered nanosheets. In addition, although the template method can prepare a cobalt sulfide-based composite photocatalyst with uniform morphology, this method requires additional removal of the template, which will increase the time and cost of the synthesis required.

## 3. Roles of Cobalt Sulfide-Based Cocatalysts in Photocatalysis

In general, the overall activity of photocatalytic reactions depends on the kinetic and thermodynamic synergy among strong light absorption capacity, charge separation rate, and surface reactivity. The interfacial chemical reaction is a key step in the process of photocatalytic reaction, which mainly involves charge transfer and redox reactions at the interface, which directly affect the efficiency of photocatalytic reaction [3,41]. Obviously, it is necessary to ensure that a large number of long-lived carriers participate in the surface reaction in order to increase the reaction rate [5]. Therefore, loading highly efficient co-catalysts on the semiconductor surface is an effective strategy to delay the recombination reaction and prolong the carrier lifetime. Cobalt sulfide-based cocatalysts, as one of the most important cocatalysts, have four critical roles in promoting the efficiency of photocatalytic reactions.

First of all, the incorporation of cobalt sulfide-based cocatalysts can facilitate the efficient separation of photoinduced carriers [6,47]. Once the cobalt sulfide-based cocatalysts are loaded on photocatalysts, numerous junctions will form due to the distinction in work function. These junctions are highly efficient contact forms between cobalt sulfide-based cocatalysts and host semiconductors, which can transfer photoexcited charges from photocatalysts to cocatalysts, thereby enabling the smooth completion of photocatalytic reactions.

Secondly, cobalt sulfide-based cocatalysts can offer adequate active sites for semiconductor photocatalysts, thereby enhancing the photocatalytic reaction potency [36]. The active sites are where the catalytic reaction proceeds and usually have low overpotential and an energy barrier for the catalytic reaction. These positions are more favorable to the catalytic reaction than other positions on the catalyst.

Thirdly, cobalt sulfide-based cocatalysts are helpful to improve the optical absorption performance of semiconductor photocatalysts [30,47]. The adsorption and activation of protons are the crucial links for enhancing the potency of the photocatalyst in the process of photocatalytic hydrogen production [30]. Cobalt sulfide with a narrow bandgap can enhance the optical absorption ability of photocatalysts by stimulating the absorption of light with a wide wavelength. In addition, cobalt sulfide can also be directly formed into a hollow structure or nanosheet array structure, which increases the specific surface area of the photocatalyst and reduces the diffusion distance of the photogenerated carrier to improve the absorption efficiency of the semiconductor.

Fourthly, the loading of cobalt sulfide-based cocatalysts can inhibit the photocorrosion of some semiconductors and enhance the stability of the photocatalysts [48]. When the cobalt sulfide-based cocatalyst with good photocatalytic activity and stability is anchored to the semiconductor, the surface reaction will be carried out on the cobalt sulfide cocatalyst, thus improving the efficiency of the photocatalytic reaction [49].

## 4. Cobalt Sulfide-Based Composite Material for Photocatalysis

### 4.1. Photocatalytic $H_2$ Production

Recently, semiconductor photocatalytic water decomposition has been improved by integrating appropriate co-catalysts. Due to the sufficient catalytic site and easy preparation, cobalt sulfide-based cocatalysts have been widely applied as co-catalysts for various semiconductors toward photocatalytic hydrogen evolution [50–53]. Fu et al. have illustrated that combining a hollow cobalt sulfide ($CoS_x$) polyhedral cocatalyst with g-$C_3N_4$ can effectively accelerate the separation of photoinduced charges in g-$C_3N_4$ and provide an abundant active site to promote redox reactions (Figure 6a) [54]. In addition, as shown in

Figure 6b, the hollow structure of the $CoS_x$ polyhedron can also allow multiple reflections of light to enhance the light collection of $g-C_3N_4$. Thus, the photocatalytic performance of the 2% $CoS_x/g-C_3N_4$ hybrids was significantly better than that of the blank $g-C_3N_4$. Obviously, the incorporation of cobalt sulfide could act as a cocatalyst to accelerate the separation and transfer of photo-generated electron-hole pairs and reduce the overpotential of the hydrogen production reaction. Qiu et al. reported that CdS nanorods loaded with $CoS_2$ nanoparticles exhibited excellent photocatalytic hydrogen production activity, which was 13 times higher than that of pristine CdS NRs samples, and the optimized $CoS_2/CdS$ NRs photocatalyst had high stability and recyclability [55].

In addition to the cobalt sulfide single component cocatalyst, multicomponent co-catalysts exhibit superior co-catalytic activity than single component cocatalysts. For example, Li et al. have reported an excellent composite photocatalyst by combining CoS with $Co(OH)_2$ on $g-C_3N_4$ to construct a dual cocatalyst [56]. The photocatalytic hydrogen production rate of the $CoS/Co (OH)_2/g-C_3N_4$ composite photocatalyst is 311 times higher than that of pure $g-C_3N_4$, which is due to the synergistic effect of the dual cocatalysts. In the dual cocatalyst system, CoS cocatalyst acts as an electron acceptor to facilitate the separation of photogenerated carriers, and $Co(OH)_2$ can also act as a conductor to diffuse photon-generated electrons. Moreover, in addition to acting as a co-catalyst, cobalt sulfide has also been reported as a semiconductor for $H_2$ production. For example, Zhang et al. used a simple hydrothermal synthesis method to in situ grow two-dimensional $ZnIn_2S_4$ on one-dimensional hollow $Co_9S_8$ nanotubes to form a $Co_9S_8/ZnIn_2S_4$ heterostructure [57]. As shown in Figure 6c, type-I heterostructures are constructed when the $Co_9S_8$ nanotubes are covered with $ZnIn_2S_4$ nanosheets. When the $Co_9S_8/ZnIn_2S_4$ composites are excited to generate electron-hole pairs, the photogenerated electrons can migrate rapidly from the CB of $ZnIn_2S_4$ to that of $Co_9S_8$. Consequently, the $Co_9S_8/ZnIn_2S_4$ heterostructure achieves a higher photocatalytic activity than pure $ZnIn_2S_4$. Apart from the aforementioned research, Table 1 summarizes other studies that have employed cobalt sulfide-based semiconductor composites for photocatalytic $H_2$ production.

**Table 1.** Cobalt sulfide-based semiconductor composites for photocatalytic $H_2$ production.

| Cocatalysts | Semiconductor | Light Source (Sacrificial Reagent) | Photocatalytic Activity ($\mu mol \cdot g^{-1} \cdot h^{-1}$) | Ref. |
|---|---|---|---|---|
| $CoS_2$ | CdS | $\lambda \geq 400$ nm (Lactic acid or $Na_2S/Na_2SO_3$) | 58,100 | [58] |
| $Co_3S_4/Co@C$ | CdS | $\lambda > 420$ nm ($Na_2S$ and $Na_2SO_3$) | 15,170 | [30] |
| $Co_4S_3$ | CdS | $\lambda \geq 420$ nm (Lactic acid) | 12,360 | [59] |
| $CdS/Co_9S_8$ | RGO | $\lambda > 420$ nm ($Na_2S$ and $Na_2SO_3$) | 4820 | [49] |
| $Co_9S_8$ | $ZnIn_2S_4$ | $\lambda > 420$ nm (TEOA) | 6250 | [60] |
| $CoS_2$ | $Zn_{0.5}Cd_{0.5}S$ | $\lambda \geq 420$ nm (L-lactic acid) | 25,150 | [48] |
| $CoS_2$ | ZnS | $\lambda > 420$ nm ($Na_2S$ and $Na_2SO_3$) | 8001 | [35] |
| $Co_3S_4$ | $g-C_3N_4$ | $\lambda \geq 400$ nm (TEOA) | 20,536.4 | [61] |
| CoS | $TiO_2$ | $\lambda \geq 400$ nm (Lactic acid) | 1945 | [47] |
| $Co_4S_3/CNFs$ | $CdIn_2S_4$ | $\lambda > 420$ nm (Lactic acid) | 25,870 | [62] |
| $Co_3S_4$ | $g-C_3N_4$ | $\lambda > 420$ nm (TEOA) | 2120 | [63] |

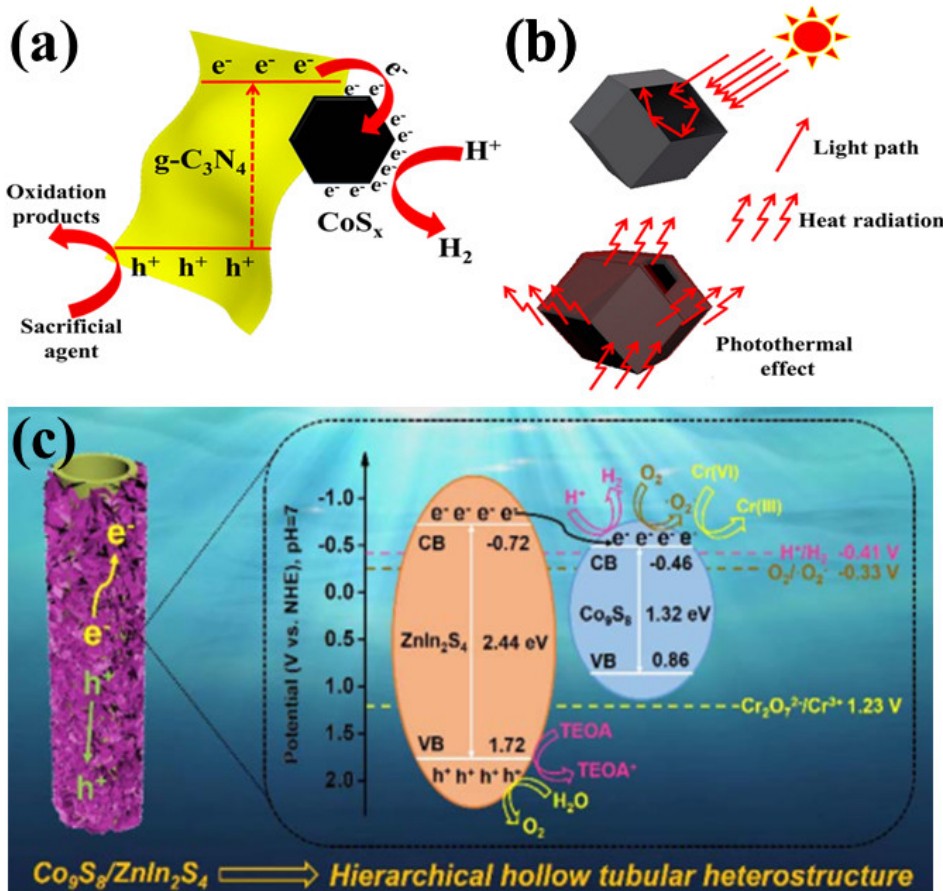

**Figure 6.** (**a**) Potential mechanism for photocatalytic $H_2$ evolution on $CoS_x/g-C_3N_4$ composite photocatalyst. (**b**) Schemes for light path and photothermal effects within the hollow $CoS_x$ polyhedron. Reprinted with permission from ref. [54]. Copyright 2018, American Chemical Society. (**c**) Schematic illustration of the fabrication process of hierarchical $Co_9S_8/ZnIn_2S_4$ tubular photocatalysts. Reprinted with permission from ref. [57]. Copyright 2020, Wiley.

### 4.2. Photocatalytic CO$_2$ Reduction

In addition to being used as a cocatalyst for photocatalytic $H_2$ production, cobalt sulfide can also be used as efficient photocatalytic for $CO_2$ reduction [64,65]. For example, Zhang et al. have composited the hollow $Co_9S_8$ nanocages with $ZnIn_2S_4$ nanosheets and CdS quantum dots to construct a ternary composite photocatalyst [29]. As shown in Figure 7a, the hollow structure of $Co_9S_8$ nanocages promotes multiple reflections of sunlight in the cavity, which enhanced the light absorption of $ZnIn_2S_4$ nanosheets and CdS quantum dots. In addition, as shown in Figure 7b, the ternary composite photocatalyst form a double Z-type heterojunction, which facilitates the separation and migration of photogenic electron hole pairs. Therefore, the photocatalytic performance of the $Co_9S_8@ZnIn_2S_4/CdS$ hybrid is obviously better than that of blank CdS and $ZnIn_2S_4$, as described in Figure 7c.

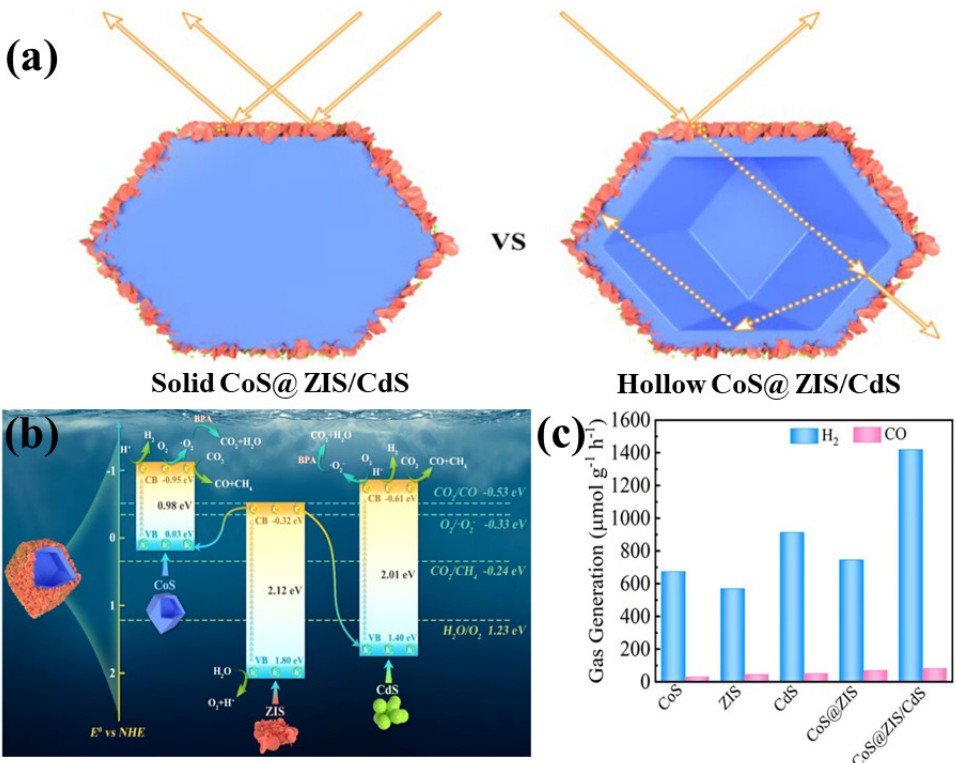

**Figure 7.** (**a**,**b**) Illustration of multiple reflections (**a**) and proposed photocatalytic mechanism over CoS@ZIS/CdS heterojunction (**b**). (**c**) Comparison of $H_2$ and CO generation over different samples. Reprinted with permission from ref. [29]. Copyright 2022, Elsevier.

Moreover, photocatalytic reduction of $CO_2$ to methanol is another ideal approach for solar energy conversion. Ma et al. have prepared carbon nitride (CN) loaded with cobalt sulfide (CS) as a cocatalyst. The optimized CS/CN photocatalyst was 2.3 times more selective for $CH_3OH$ than CN [66]. It was confirmed that the introduction of cobalt sulfide can improve the selectivity of $CH_3OH$. The cobalt sulfide not only provides the $H_2O$ oxidation center but also can significantly weaken the overpotential of the $H_2O$ oxidation half reaction, thus effectively avoiding the formation of strongly oxidized radicals.

Furthermore, Wang et al. have reported hierarchical $FeCoS_2$-$CoS_2$ double-shelled nanotubes as a composite photocatalyst for $CO_2$ reduction [67]. As shown in Figure 8a, $FeCoS_2$-$CoS_2$ composites can be obtained after ion-exchange reactions and sulfidation reactions with MIL-88A as precursors. As shown in Figure 8b, $FeCoS_2$-$CoS_2$ composites present a uniform hierarchical nanosheet structure. When the $Ru(bpy)_3^{2+}$ is used as the photosensitizer, the optimal $FeCoS_2$-$CoS_2$ composite shows excellent photocatalytic activity with a CO generation rate of 28.1 µmol $h^{-1}$, which is better than the individuals of $FeCoS_2$ and $CoS_2$ and their physical mixtures sample (Figure 8c). As illustrated in Figure 8d, the unique hierarchical nanosheet structure reduces diffusion length and enhances scattering in the cavity, which inhibits electron-hole recombination and exposes active sites for redox reactions, thus improving the photocatalytic activity of the $FeCoS_2$-$CoS_2$ composite.

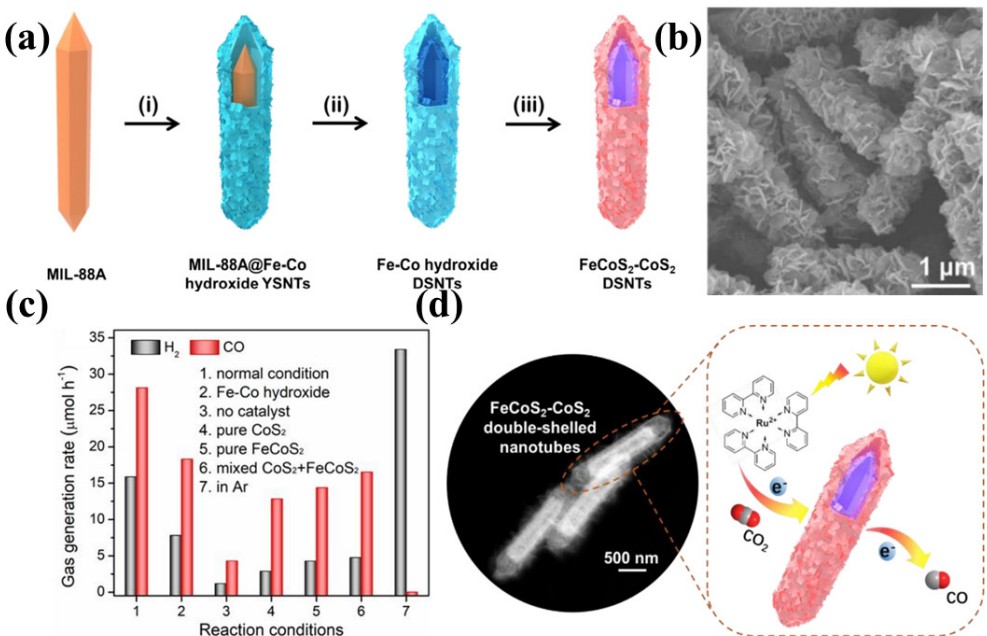

**Figure 8.** (**a**) Illustration of the synthetic process for hierarchical FeCoS$_2$−CoS$_2$. (**b**) FESEM images of one FeCoS$_2$−CoS$_2$. (**c**) Photoreduction of CO$_2$ under different reaction conditions. (**d**) Schematic diagram of CO$_2$ photoreduction over FeCoS$_2$-CoS$_2$. Reprinted with permission from ref. [67]. Copyright 2020, Wiley.

### 4.3. Photocatalytic Nitrogen Fixation

Neither humans nor the earth's ecosystem can survive without the ability to synthesize ammonia [68]. The production of this foundation sustaining life on earth is based on both industrial and biological fixation levels of $200 \times 10^6$ tons per year [69]. At present, nitrogen fixation is principally carried out in three ways: (i) biological nitrogen fixation. Some micro-organisms, such as nitrogen-fixing bacteria, use their own nitrogenase to fix N$_2$ molecules for biological nitrogen fixation; (ii) high-energy nitrogen fixation in geochemical processes, such as lightning; (iii) the energy-intensive Haber–Bosch method for industrial nitrogen fixation. However, biological and geochemical nitrogen fixation solely account for a tiny fraction of the fixed nitrogen supply. The Haber–Bosch process, which uses N$_2$ and H$_2$ as sources and iron-based compounds as the main material, is currently the main route for the synthesis of industrial ammonia. Nevertheless, this process requires a great deal of energy input while generating large emissions of by-products (such as carbon dioxide), which may cause environmental hazards. Hence, developing high-selectivity photocatalysts for nitrogen-reducing ammonia is challenging and interesting research [69]. Recently, Yuan et al. have demonstrated that loading Ru/CoS$_x$ to g-C$_3$N$_4$ nanosheets can effectively activate N$_2$ molecules and facilitate the separation of light-induced electron-hole pairs in g-C$_3$N$_4$ [70]. As shown in Figure 9a, in comparison with pure CN, Ru-Vs-CoS/CN shows obviously enhanced photocatalytic activity, reaching 1.28% apparent quantum efficiency at 400 nm and 0.042% solar-to-ammonia efficiency. The excellent nitrogen reduction reaction performance is attributed to the fact that the sulfur vacancies in CoS$_x$ can effectively promote the selective chemisorption of N$_2$ molecules. In addition, an N$_2$ molecule is bridged against the side-on Ru-Co center by the undercoordination of Ru and Co atoms at the Ru/CoS$_x$ interface. Furthermore, as shown in Figure 9b, the plasmonic Ru/CoS$_x$ interface enhances light absorption to generate energetic charge-carriers, accelerates charge separation and transfer, and therefore kinetically facilitates the fixation of N$_2$. This confirms that the presence of vacancies on the surface of cobalt sulfide-based nanomaterials exhibits excellent photocatalytic NRR performance, as it can modify the electronic structure, decrease the coordination number of surface atoms, facilitate the formation of dangling bonds, and greatly promote the formation of N$_2$ chemisorption

and activation. The $N_2$-fixation mechanism outlined in Figure 9c indicates the hydrogen evolution reaction (HER) on Ru occurs easily due to its good free energy of hydrogen production ($-0.07$ eV). Meanwhile, the active hydrogen adsorption on Co and desorption on S limit the hydrogen evolution reaction (HER) on Ru.

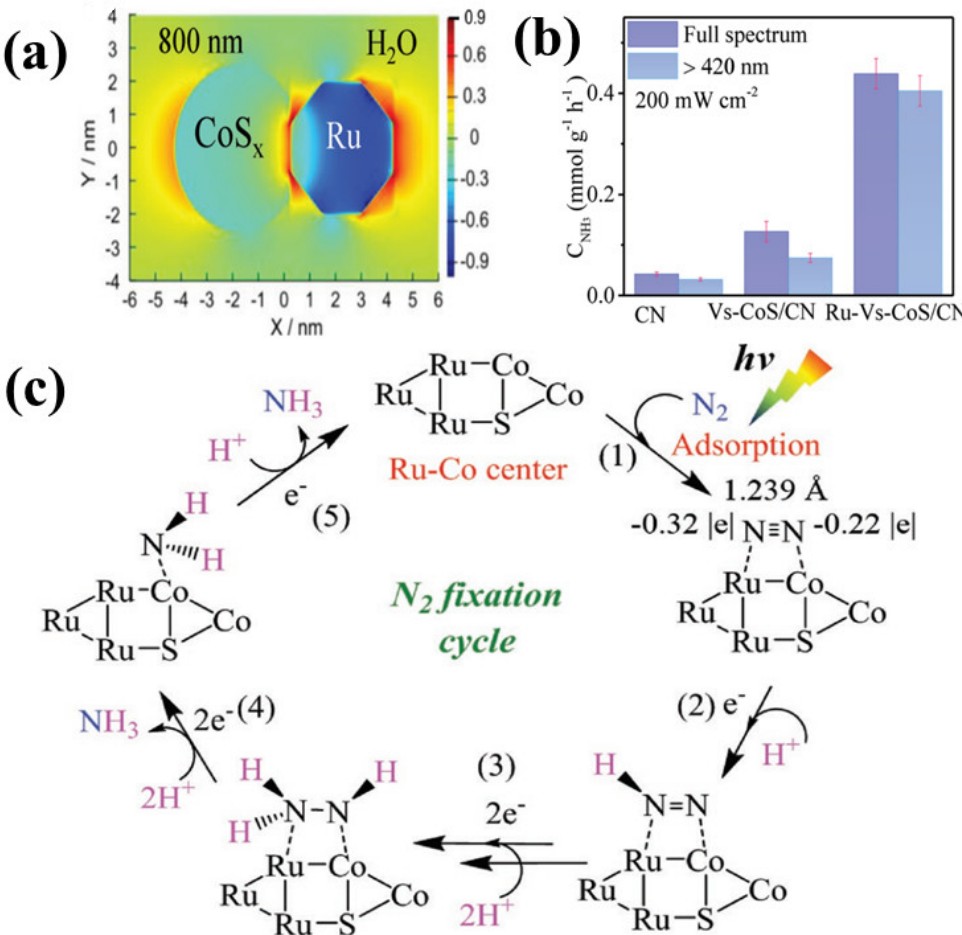

**Figure 9.** (**a**) Photocatalytic $NH_3$ production rates over different samples (**b**) FDTD electric field distribution observed from the *z*-axis (parallel to incident light) at $Ru/CoS_x$ nanoparticles (800 nm irradiation). (**c**) Proposed photocatalytic $N_2RR$ pathway on $Ru-Vs-CoS/CN$. Reprinted with permission from ref. [70]. Copyright 2020, Wiley.

### 4.4. Photocatalytic Degradation

Recent research shows that cobalt sulfide-based materials, such as CoS, $CoS_2$, and $Co_3S_4$, are important candidate catalysts for photocatalytic organic pollutants degradation [71–74]. For instance, $Co_{2.67}S_4$ shows excellent photocatalytic degradation efficiency of methylene blue (MB) under UV, visible, and near-infrared irradiation [75]. As shown in Figure 10a, the valence state change of cobalt ions effectively separates electrons from holes and accelerates electron transfer, thus enhancing the activity of photocatalytic degradation. In addition to single cobalt-based sulfide materials, cobalt sulfide, as a co-catalyst, can be combined with host semiconductors for photocatalytic degradation. For example, Tang et al. have designed a two-dimensional CoS/BiOBr heterojunction, which shows a 5.3-fold higher degradation rate as compared to pure BiOBr (Figure 10b) [76]. As shown in Figure 10c,d, when the BiOBr and the CoS combine to construct the CoS/BiOBr heterojunction photocatalyst, the electrons on the CB of the CoS can be easily transferred to the CB of the BiOBr. In addition, the VB of BiOBr can oxidize glyphosate directly, producing small molecules or ions ($PO_4^{3-}$, etc.). Simultaneously, some holes also migrate from BiOBr to

CoS, leading to effective photogenerated charge carrier separation and thereby boosting the photocatalytic performance of the CoS/BiOBr composite.

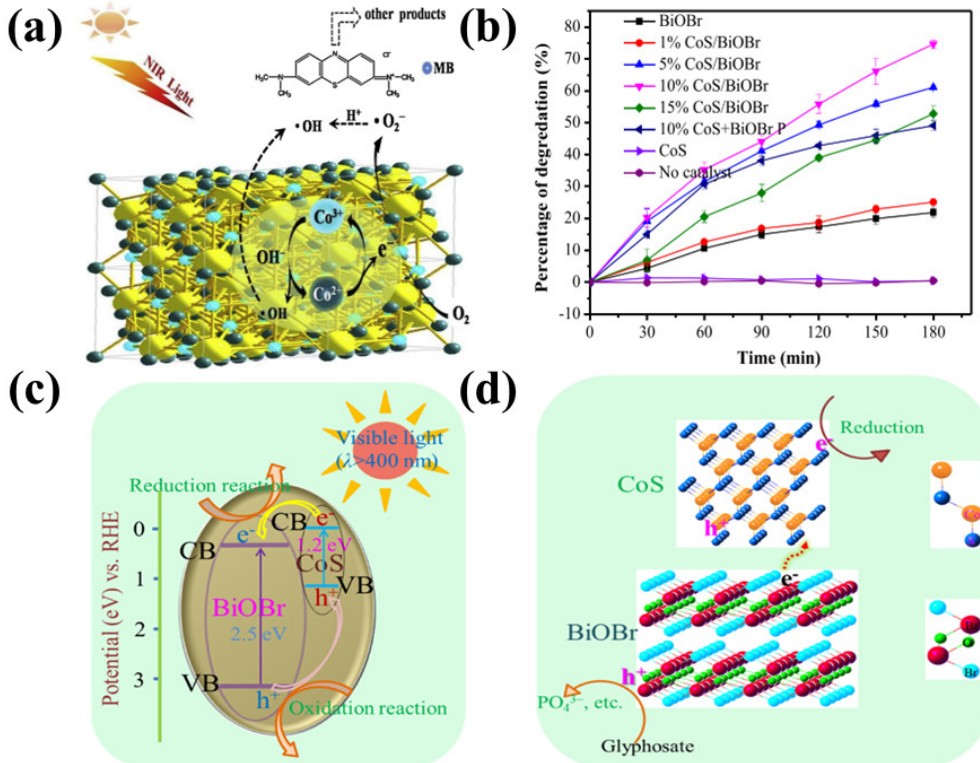

**Figure 10.** (**a**) Illustrative diagram of the $Co_{2.67}S_4$ system under NIR light irradiation. Copyright 2017, Elsevier. (**b**) Photocatalytic degradation of glyphosate. (**c**) A schematic illustration of the photogenerated carrier transfer process and (**d**) photocatalytic degradation process over CoS/BiOBr. Reprinted with permission from ref. [76]. Copyright 2021, Elsevier.

Moreover, Zhang et al. have covered uniformly $MoS_2$ nanosheets on $CoS_2$ nanoparticles to construct $CoS_2/MoS_2$-nitrogen-doped graphene aerogels for photocatalytic organic pollutants degradation [77]. When $MoS_2$ is combined with $CoS_2$, the band gap of $MoS_2$ can be narrowed and the optical response range can be expanded. At the same time, $CoS_2$ can effectively accelerate the charge separation and increase the surface-active sites. Taking advantage of these advantages, the optimized three-dimensional $CoS_2/MoS_2$-nitrogen-doped graphene aerogel photocatalyst can degrade pollutants up to 97.1% within 60 minimums and still maintain 95.1% after three cycles. Apart from the aforementioned research, Table 2 summarizes other studies that have employed cobalt sulfide-based composites for photocatalytic organic pollutant degradation.

**Table 2.** Cobalt sulfide-based semiconductor composites for photocatalytic degradation.

| Catalysts | Conditions | Catalyst Amount (mg) | Dye/Concentration | %Degradation/Time (min) | Ref. |
|---|---|---|---|---|---|
| Pg-$C_3N_4$ /$Co_3O_4$/CoS) | different pH (pH = 3, 5, 7, 9, 11) 500 W Xe lamp | 5 | BPF/30 mg·$L^{-1}$ | 99/50 | [72] |
| CoS-TEA | 300 W Xe lamp | 20 | RhB/10 mg·$L^{-1}$ | 97.34/80 | [78] |
| CoS-rGO | pH = 5, sunlight with light intensity of ~680 W/m | 5 | CR/10 mg·$mL^{-1}$ | 88.03/40 | [79] |

**Table 2.** *Cont.*

| Catalysts | Conditions | Catalyst Amount (mg) | Dye/Concentration | %Degradation/Time (min) | Ref. |
|---|---|---|---|---|---|
| CoS-rGO/PMS | different operating conditions at room temperature | 25 | RhB/14 mg·L$^{-1}$ | 95/8 | [80] |
| Co$_3$S$_4$-SnO$_2$/PVPCS | 25 W UV lamp (UV lamp: 5.5 cm, and light intensity: at 3.0 mW·cm$^{-2}$) | 10 | LDC/10 mg·mL$^{-1}$ | 98.72/30 | [81] |
| CoS NS | Neutral pH, 200 W tungsten lamp | 5 | MB/20 mg·L$^{-1}$<br>RhB/20 mg·L$^{-1}$<br>CV/20 mg·L$^{-1}$<br>NB/20 mg·L$^{-1}$ | 99.8/10<br>99.5/45<br>99.4/3<br>99.8/5 | [82] |
| CdS/N-CoS$_x$ | 300 W Xe lamp | 10 | Cr(VI)/10 mg·mL$^{-1}$ | 100/25 | [83] |
| Graphite/Cobalt Sulfide/PANI composite | Magnetic Agitation under dark and Visible light (15 watt) | 25 | CR/25 mg·mL$^{-1}$ | 99.55/120 | [84] |
| CoS$_2$-CeO$_2$/CSCS | under UV light irradiation | 20 | 4NP/10 mg·L$^{-1}$ | 95.42/60 | [85] |

## 5. Conclusions and Perspectives

In this review, we have summarized recent progress in cobalt sulfide syntheses, especially morphological and temperature-dependent design guidelines, and their applications in photocatalytic hydrogen production, CO$_2$ reduction, nitrogen fixation, and degradation pollutant. In spite of the significant progress made to date, some challenges and opportunities for further advancement in this research field are presented as follows:

(1) Nowadays, cobalt sulfide is regarded as an inexpensive, easily synthesized, and efficient photocatalyst. However, cobalt sulfide is much less stable than the catalysts required for practical applications. Therefore, more efforts need to be made to enhance the stability of cobalt sulfide;

(2) To date, there are almost no practical synthetic methods for cobalt sulfide-based composites that are available for mass production to meet real-life applications. Therefore, the development of industrial-scale production methods with stable, efficient, and low-cost cobalt sulfide-based composites is significant;

(3) Since sacrificial agents are inevitably used for current photocatalytic reactions, this causes serious problem of increased reaction costs and waste of reaction energy. In addition, the enhancement of photocatalytic activity is mainly determined by the consumption degree and survival time of its photosynthetic holes or electrons. In this regard, the combination of H$_2$ production, the reduction of CO$_2$, and N$_2$ fixation with oxidative organic synthesis in a photosynthetic reaction is a feasible method for avoiding the use of sacrificial agents;

(4) Many problems still need to be addressed in further development. For example, studies on the active sites, and charge carrier dynamics of cobalt sulfide catalysts are still in their infancy. In addition, the mechanism of cobalt sulfide as a photocatalytic catalyst also deserves further investigation. Therefore, it is very necessary to conduct more thorough and systematic studies of these problems, both theoretically and experimentally. Notably, in situ characterization techniques are capable of detecting the change of structure within the cobalt sulfide group in real time, which requires more effort to develop.

**Author Contributions:** Conceptualization, H.-Q.C. and K.-Q.L.; software, J.-G.H., Y.W. and K.Y.; formal analysis, W.-Y.H., J.-L.Z. and T.D.; resources, K.Y., W.-Y.H. and K.-Q.L.; data curation, H.-Q.C. and J.-G.H.; writing—original draft preparation, H.-Q.C.; writing—review and editing, K.-Q.L.; supervision, K.-Q.L. All authors have read and agreed to the published version of the manuscript.

**Funding:** This study is financially supported by Jiangxi University of Science and Technology students' innovation and entrepreneurship training program (Preparation of graphene aerogel/ semiconductor composite photocatalytic materials and their performance research, 202210407022), the Jiangxi Provincial Natural Science Foundation (20212BAB213016, 20224BAB203018, 20224ACB213010, 204302600031), the Postdoctoral Research Projects of Jiangxi Province (2021RC11, 204302600031), the Jiangxi Province "Double Thousand Plan", the high-level talent research launch project of JXUST (205200100518), the National Natural Science Foundation of China (21962006), the Jiangxi Provincial Academic and Technical Leaders Training Program—Young Talents (20204BCJL23037), the Program of Qingjiang Excellent Young Talents, JXUST(JXUSTQJBJ2020005), the Ganzhou Young Talents Program of Jiangxi Province (204301000111), and the Jiangxi Provincial Key Laboratory of Functional Molecular Materials Chemistry (20212BCD42018).

**Data Availability Statement:** The original contributions presented in the study are included in the article; further inquiries can be directed to the corresponding author.

**Conflicts of Interest:** The authors declare no conflict of interest.

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
