# Peer review of "Recent Developments and Perspectives of Cobalt Sulfide-Based Composite Materials in Photocatalysis"

_catalysts, doi:10.3390/catal13030544_

Round 1

Reviewer 1 Report

Manuscript ID: catalysts-2196964

The manuscript entitled, “Recent Developments and Perspectives of Cobalt Sulfide-Based Composite Materials in Photocatalysis” is a good work. Authors nicely review the CoS base composites used for photocatalysis. Although topic is interesting but much work is required for possible publication in worthy journal Catalyst. I had selected major revision to improve the quality of this article and to enhance the novelty and better readership. My comments are,

1.      In abstract add more quantitative data.

2.      Characterization should be discussed in more detail and relevant figures should be cited

3.      Very few data/article reviewed in table 1 and 2. More data should be added. Similarly for section 4.2. photocatalytic CO2 reduction and 4.3. photocatalytic nitrogen fixation similar tale like table 1 should be added.

4.      In revised manuscript cite these latest articles for photo-degradation of contaminants, Chemical Physics Letters 805 (2022) 139939, https://doi.org/10.1080/03067319.2022.2032014, https://doi.org/10.1080/24701556.2021.1980021.

5.      Many articles are reported on photocatalysis with Cos composites. Authors have reviewed very limited literature. For more in-depth analysis more articles should be reviwed and relevant figures from characterization and application should be added in revised manuscript.

6.      The whole manuscript must be cross-checked thoroughly for English editing, grammatical, spelling mistakes, and syntax errors.

Reviewer 2 Report

Abstract

Please revise it and make it more objective oriented

Introduction

The authors presented Cobalt-sulphide as suitable photolytic candidate. I think they should little describe about the role of sulfur as well. As sulfur is supposed to be hazardous material.

Some headings and sub headings needs to be corrected in formatting errors. Like 4.4

I would suggest if the authors would add role of composite on band gap towards the efficiency of photo catalytic degradation

Please double check spelling and grammar

Reviewer 3 Report

The present review manuscript entitled “Recent Developments and Perspectives of Cobalt Sulfide-Based Composite Materials in Photocatalysis” by Chen et al., describes the strategies for preparing cobalt sulfide-based composite materials. Furthermore, the roles of cobalt sulfide-based cocatalysts in photocatalysis have been discussed and finally, the development prospects and challenges of cobalt sulfide-based photocatalysts in the energy research field are discussed. The authors report an interesting review. The objective and justification of the work are clear and I congratulate the authors for their good work. The review is more a comment than a review in the traditional sense and hence I would strongly advise that it would be published as such. However, certain Minor issues are detailed below to improve the quality of the manuscript.

I advise the authors to take the following points into account while revising their manuscript.

Comment 1: There are some typographical errors in the manuscript text, so the authors need to correct them in the revised manuscript.

Comment 2: English needs to be a little improved, as there are some misused conjunctions and technical flaws to correct in the manuscript.

Comment 3: Since the introduction is too short, to improve the introduction section the authors need to cite and discuss some more recent references in the introduction section to strengthen the section.

Comment 4: please check and revise the keywords.

Comment 5: Please enhance the novelty statement at the end of the introduction section. Please add why the study is important and what are the outcomes of the study.

Comment 6: Include the structured graphical abstract in the revised manuscript to attain a broad readership.

Comment 7: The homogeneity of the reference section needs to be maintained. In reference no. 34, the journal name is written in full form and some references are in abbreviations form. So please check and revise accordingly to the journal instructions.

Round 2

Reviewer 1 Report

Accept in present from